# The Effect of Dielectric Polarization Rate Difference of Filler and Matrix on the Electrorheological Responses of Poly(ionic liquid)/Polyaniline Composite Particles

**DOI:** 10.3390/polym12030703

**Published:** 2020-03-22

**Authors:** Chen Zheng, Qi Lei, Jia Zhao, Xiaopeng Zhao, Jianbo Yin

**Affiliations:** Smart Materials Laboratory, Department of Applied Physics, Northwestern Polytechnical University, Xi’an 710129, China; zhengchen2603@mail.nwpu.edu.cn (C.Z.); leiqi@mail.nwpu.edu.cn (Q.L.); zhaojia11@mail.nwpu.edu.cn (J.Z.); xpzhao@nwpu.edu.cn (X.Z.)

**Keywords:** composite particles, dielectric polarization rate, electrorheological responsive polymer

## Abstract

By using different conductivity of polyaniline as filler, a kind of poly(ionic liquid)/polyaniline composite particles was synthesized to investigate the influence of dielectric polarization rate difference between filler and matrix on the electrorheological response and flow stability of composite-based electrorheological fluids under simultaneous effect of shear and electric fields. The composite particles were prepared by a post ion-exchange procedure and then treated by ammonia or hydrazine to obtain different conductivity of polyaniline. Their electrorheological response was measured by dispersing these composite particles in insulating carrier liquid under electric fields. It showed that the composite particles treated by ammonia had the strongest electrorheological response and most stable flow behavior in a broad shear rate region from 0.5 s^−1^ to 1000 s^−1^. By using dielectric spectroscopy, it found that the enhanced electrorheological response with stable flow depended on the matching degree of the dielectric polarization rates between poly(ionic liquid) matrix and polyaniline filler. The closer their polarization rates are, the more stable the flow curves are. These results are helpful to design optimal composite-based electrorheological materials with enhanced and stable ER performance.

## 1. Introduction

Polymer composites, consisting of at least two components, a polymer matrix and a filler, can exhibit enhanced physicochemical properties compared to neat polymers. Besides as structured materials, they have also been frequently used as functional and smart materials, such as high thermal conductivity materials, electromagnetic interface shielding or absorption materials, shape memory materials, field-responsive materials [1,2,3,4,5]. In these applications, the performance of the whole composites depends on not only the single properties of matrix or filler but also the matching degree of properties of matrix and filler. For example, the good elastic matching between matrix and filler can decrease the stress shielding behavior of composites [6]. The impedance matching of matrix and filler has an influence on the microwave absorption or shielding performance of composite [7]. Therefore, research on the property matching degree between matrix and filler is of great use to design polymer composites with optimal performance.

Besides use with a bulk form, polymer composite particles have also been used as the active dispersed phase in some functional and smart fluids, such as polishing fluid, electrorheological (ER) fluid, and magnetorheological (MR) fluids [8,9,10]. Electrorheological fluid is an electro-responsive suspension, which is composed of polarizable particles and insulating liquid medium [11]. Without electric fields, the particles are freely dispersed. With external electric fields applied, the particles are polarized and attracted to form particle chains along electric fields. This orientated structure can enhance the viscosity of ER fluid and even make ER fluid transform from liquid-like state into solid-like state, the so-called ER effect [12,13]. Thanks to this tunable ER effect, ER fluid has many potential uses as an electrical-mechanical interface to develop smart devices, such as smart dampers, soft robotics, and microfluidic rectifier [14,15,16,17]. To promote real applications, ER fluids containing various kinds of particles have been developed, such as semiconducting polymer particles, inorganic particles, polyelectrolyte particles, and carbonaceous particles [18,19,20,21,22]. In addition, the ER fluid based on polymer composite particles has also been frequently studied because they often have enhanced properties, such as decreased low density, enhanced thermal stability, and ER activity [23,24]. For example, the ER fluid with polyaniline (PANI)/graphene composite particles shows increased yield stress due to graphene-induced enhancement of electric polarization [25]. The ER fluid with PANI/montmorillonite composite particles shows not only enhanced temperature stability but also good dispersion stability [26]. Recently, some poly(ionic liquid) (PIL)-based composite ER particles have also been developed in order to enhance the performance of some PIL-based ER materials, a new type of anhydrous polyelectrolyte-based ER system [27,28]. For example, Chen et al. used a multi-layer coating technique to prepare graphene oxide/polypyrrole/PIL composite ER particles showing enhanced yield stress. Zhao et al. used a Pickering emulsion polymerization to prepare PIL/nano-SiO_2_ composite ER particles showing enhanced temperature stability [29,30]. However, these studies have focused on yield stress or thermal stability.

In many applications, besides yield stress, the flow stability of ER fluid under the simultaneous effects of shear and electric fields is also important. It has proposed that the flow stability is closely related to the dielectric polarization rate of dispersed particles [31,32]. A polarization rate that is too slow cannot provide enough time with particles to rebuild particle chains at a high shear rate, while polarization rate that is too fast easily leads to the interparticle repulsion as soon as the direction of particle chains is not consistent with the direction of electric fields under shearing. It is proposed that the polarization rate located in the frequency range of 10^2^–10^5^ Hz or 6.28 × 10^2^–6.28 × 10^5^ rad/s may be proper to fulfill a stable flow behavior under DC electric fields. However, this proposal is concluded based on single component ER particles. In recent PIL-based ER fluid, for example, the stable flow has been obtained when employing an appreciate pendant group to endorse PIL particles with suitable polarization rates [33]. In addition, changing the crosslinking degree or mesh size of crosslinking network has also adjusted the polarization rate and the resulting flow stability [34]. Whether this is the proposal available for composite ER particle systems, in particular for composite particles with multi-level polarization rates? The answer to this question is valuable for the choice and design of the composite-based ER materials with optimized performance.

In this work, we prepared a kind of composite ER particles with PIL as matrix and PANI as filler and investigated the effect of dielectric polarization rate difference between matrix and filler on the ER response and flow stability of ER fluid under simultaneous effect of shear and electric fields. We prepared PIL/PANI composite particles by ion-exchange and then treated by ammonia or hydrazine to obtain different forms of PANI filler, such as emeraldine salt, emeraldine base, and leucoemeraldine base [35,36]. Because of their significantly different conductivities, we can adjust the difference in polarization rate between filler and matrix. The ER response was measured by a rheometer under electric fields. The dielectric polarization was analyzed by dielectric relaxation spectroscopy. It showed that the variations of the flow behavior of these ER fluids were consistent with the matching degree of dielectric polarization rate between PIL matrix and PANI filler. The closer their polarization rates were, the more stable the flow curve was in a broad shear rate region.

## 2. Materials and Methods

### 2.1. Materials

(Vinylbenzyl)trimethylammonium chloride ([VBTMA]Cl, 99%) was purchased from Sigma-Aldrich, St. Louis, MO, USA. Potassium hexafluorophosphate (K[PF_6_], 99%) was purchased from J&K Chemical, Beijing, China. 2,2′-azobis(isoheptonitrile) (AIBN), aniline, ammonia (NH_3_·H_2_O, 25%), hydrazine hydrate (N_2_H_4_·H_2_O, 80%), hydrochloric acid (HCl, 36%), and silver nitrate (AgNO_3_, 99%) were purchased from Sinopharm Chemical Reagent Co., Ltd., Shanghai, China. Ammonium persulfate (APS) was purchased from Tianli Chemical Reagent Co., Ltd., Tianjin, China. Dimethyl silicone oil (KF-96) was purchased from Shin-Etsu Chemical Co., Ltd., Tokyo, Japan. All these reagents were used as received except that AIBN was purified by recrystallization in ethanol.

### 2.2. Preparation of PIL/PANI Composite Particles

PIL/PANI composite particles were prepared by the modified method of Marcilla et al. [37]. The preparation details of P[VBTMA][PF_6_] particles and P[VBTMA][PF_6_]/PANI composite particles refer to our previous work [38]. The difference is that a spot of HCl was added into the reaction system to dissolve aniline at the outset. Then, a silver nitrate solution was used to detect the absence of chlorine ion to verify the ion exchange completely. What is more, in order to obtain the different forms of PANI, the product was equally divided into three parts for different post-processing. The one part was dipped in NH_3_·H_2_O (50 mL, 3 wt%) for 5 min, the second part was reduced by N_2_H_4_·H_2_O (20 mL) at 95 °C for 6 h, and the third part accepted no special treatment. After washing with deionized water and vacuum drying, the different polarization rates of PIL/PANI composite particles were obtained.

To facilitate characterizing the structure of these composite particles, the PIL/SiO_2_ composite particles were prepared by the same method.

### 2.3. Preparation of ER Fluids

The PIL, PIL/PANI (HCl), PIL/PANI (NH_3_·H_2_O), and PIL/PANI (N_2_H_4_·H_2_O) particles were further washed with ultrapure water, and dried in vacuum at 100 °C for 48 h. Then, the dry samples were added into insulating dimethyl silicone oil whose kinetic viscosity is 50 cSt at 25 °C and ground to uniform fluids. The particle concentration of fluids is 20 vol%.

### 2.4. Characterization

The morphology of samples was observed by scanning electron microscopy (SEM, Hitachi TM3000, Tokyo, Japan) and high-resolution transmission electron microscopy (HRTEM, FEI Talos F200X, Hillsboro, OR, USA). The operating voltage of SEM is 15 kV and that of HRTEM is 200 kV. The element composition of the composite particle was characterized by the HRTEM equipped with energy dispersive X-ray spectroscopy (EDS, FEI Talos F200X’s own, Hillsboro, OR, USA). The chemical groups of particles, which were mixed with KBr and pressed into platelets, were detected by the Fourier transform infrared spectrum (FT-IR, JASCO FT/IR-470 Plus, Tokyo, Japan). The absorption spectrum was measured by using Ultraviolet and visible spectrophotometer (UV-vis, Hitachi U-4100, Tokyo, Japan).

### 2.5. Rheological Measurements

The ER structure formed under an electric field was observed by an optical microscope (Nikon ALPHAPHOT-2, Tokyo, Japan). The electric field strength for the suspension in the gap was around 2 kV/mm to arrange well-balanced column-like ER structures.

The ER response of fluids was tested by a stress-controlled rheometer (Thermal-Haake RS600, Karlsruhe, Germany) with a parallel plate system and an oil bath system. The electric field through two plates was applied by a DC high-voltage generator (WYZ-010, Beijing, China), and the electrode gap was 1.0 mm. The flow curves of the shear stress as a function of shear rate were tested by the controlled shear rate mode within 0.1–1000 s^−1^, and the test temperature range was from 25 °C to 120 °C. Before every measure started, the suspension was pre-sheared for 60 s at 100 s^−1^ to remove residual ER structures from the last test and then the electric field was applied for 10 s to form equilibrated gap-spanning ER structures.

### 2.6. Dielectric Spectroscopy Measurements

The dielectric properties of these ER fluids were measured by an impedance analyzer (HP 4284A, Santa Clara, CA, USA) with a liquid measuring fixture (HP 16452A, Santa Clara, CA, USA). The measurement frequency range was 20–10^6^ Hz. The bias voltage was 1 V during measurements, which could not cause fibrous-like structures establishment. Hence, the polarization characteristics of different particles could be well compared. Before every temperature measurement, the liquid measuring fixture was equilibrated for at least 5 min to achieve thermal stabilization within 0.2 °C by using a constant temperature oil bath.

## 3. Results and Discussion

### 3.1. Morphology and Structure of PIL/PANI Composite Particles

Figure 1A–D exhibits the SEM images of PIL, PIL/PANI(HCl), PIL/PANI(NH_3_), and PIL/PANI(N_2_H_4_) particles. It can be seen that all the particles are irregular with a similar size distribution of 4–10 μm owing to the same synthesis process. Furthermore, there is no PANI structure that looks like broccoli being observed which indicates that the PANI is wrapped by PIL during the ion exchange process. It is noted that the surface of PIL/PANI(N_2_H_4_) particles is obviously different from the others due to the high-temperature reaction by the reduction of N_2_H_4_·H_2_O.

For further analysis, Figure 2A shows the TEM image of an ultrathin cross-section of the PIL/PANI particle. An overlay of nitrogen belonging to the poly[VBTMA] cation part and PANI, and phosphorus belonging to the [PF_6_] anion part EDS maps is shown in Figure 2E. It can be clearly found that the distribution of elements is not homogeneous. The phosphorus tends to distribute on the edge of particles, which indicates the unsymmetrical structure by the post ion-exchange procedure. This can be confirmed with our previous work, in which we proved the PIL/PANI structure by comparing the electrical response with PANI/PIL mixture [38]. However, it is still difficult to distinguish the distribution of PANI due to that both PIL and PANI have nitrogen element. Therefore, the same mass fraction of SiO_2_ is applied to take the place of PANI by using the same preparation method. As shown in Figure 2G–L, it can be clearly seen that the SiO_2_ particles are inlaid inside the PIL particle. Hence, it can be inferred that the distribution of PANI is a similar situation. In addition, the structure can be also verified by the optical photographs in Figure 4, where PIL/PANI are dark particles capsulated by a transparent shell.

Figure 3A shows the FTIR spectra of neat PIL, PIL/PANI(HCl), PIL/PANI(NH_3_), and PIL/PANI(N_2_H_4_) particles. All the lines show the characteristic peaks of PIL, which peaks designated to the poly[VBTMA] cation part are at 3054 cm^−1^ (CH_3_), 2926 cm^−1^ (CH_2_), 1625 cm^−1^, and 1491 cm^−1^ (C=C in benzene ring), and the peaks designated to the [PF_6_] anion part at 838 cm^−1^ and 558 cm^−1^ (P–F). As for PIL/PANI composite particles, the peaks corresponding to PANI(HCl) are at 1585 cm^−1^ (C=C in quinoid), 1477 cm^−1^ (C=C in benzene ring), 1300 cm^−1^ (C–N), and 1126 cm^−1^ (electronic-like absorption of N=Q=N (Q representing the quinoid)). After treated by NH_3_·H_2_O or N_2_H_4_·H_2_O, the peaks ascribed to the C=C stretching of quinoid, C=C stretching of benzene ring, and electronic-like absorption of N=Q=N show an obvious blue shift to 1596 cm^−1^, 1512 cm^−1^, and 1170 cm^−1^. In addition, the quinonoid structure can be reduced by N_2_H_4_·H_2_O to benzenoid structure, which leads to that the intensity ratio of the peak at 1596 cm^−1^ then to the peak at 1512 cm^−1^ decreasing. These changes in the PANI structure can be also confirmed by the UV-vis spectra (see Figure 3B). After dedoping by NH_3_·H_2_O, the absorption peaks also display a slight blue shift in the C=C stretching vibration of benzenoid and quinonoid rings from 322/615 nm to 315/600 nm. After reducing by N_2_H_4_·H_2_O, the intensity of absorption peak attributed to benzenoid ring increases while that attributed to quinonoid ring decreases. These changes in the PANI chemical structure will bring the differences in electrical properties for PANI filler, such as conductivity, polarization, and energy gap [39,40].

### 3.2. Electrorheological Properties

Figure 4 displays the optical photographs of these ER fluids between two electrodes. Without the electric field, these particles disperse in the silicone oil randomly. The PIL particles are white transparent, while the PIL/PANI composite particles are black. These composite particles look uniform without white impurities, which indicates that the PANI is trapped by PIL in the counterion conversion process. With an electric field applied, the randomly distributed polarizable particles quickly attract each other and shape into gap-spanning column-like structures attributed to the interparticle electrostatic interaction. This column-like structure in a direction perpendicular to shear flow will increase the viscosity of the fluids and even generate yield stress leading to a liquid-solid transition, the so-called ER response [12]. We note that there are some differences in the column-like structure between PIL and PIL/PANI particles. For PIL, the columns seem to be not asymmetric, and more particles tend to aggregate near the negative electrode. While for PIL/PANI, the columns seem to be more uniform except Figure 4D. This phenomenon has been discussed in our previous work in detail, so that is not repeated any more in this study [38].

Figure 5 shows the variation of shear stress with shear rate for the PIL/PANI ER fluids at different electric fields. Without an electric field applied, all these fluids exhibit a shear thinning phenomenon with a low viscous state. The off-field viscosities of these fluids are very close, which are around 0.17 Pa·s at 1000 s^−1^, owing to the similar particle size and morphology for all these particles. This can also prove that the PANI is surrounded by PIL for composite particles. With the electric field applied, all of the fluids show a conspicuous increase in shear stress and possess large yield stress like a Bingham fluid. The increase in yield stress can be adjustable with electric field strength because the polarization intensity of the dispersed particles can be enhanced with the external electric field increasing, leading to the stronger formation level of column-like ER structures [41]. What is more, there are many obvious differences among the flow curves of these suspensions. Compared to the flow curves of the ER fluid of PIL particles, those of PIL/PANI(HCl) particles exhibit a wave trough in high shear rate region at high electric field strength, those of PIL/PANI(NH_3_) particles possess much larger shear stress, and those of PIL/PANI(N_2_H_4_) particles look like continuous upward slopes. It has been known that the flow behavior of ER fluid is closely connected with the rupture and recombination of field-induced column ER structures under the electric field and shear field simultaneously. The stable flow curve in a broad shear rate region requires that the dispersed polarizable particles possess a suitable polarization rate to rebuild column structures effectively. Hence, the structural state of PANI filler has a distinct influence on the electric response of composite particles leading to these differences in flow behavior of the four suspensions.

Figure 6 plots the electric field strength dependence of the static yield stress (τ_s_) and the dynamic yield stress (τ_d_) for these suspensions. The τ_s_ characterizes the solidified level of ER fluid before flowing. While the τ_d_ represents the strength of suspension in the flow regime [19,42], which can be obtained by extrapolating the fitting lines of flow curves with the Bingham model (see Figure 5). It is needed to point out that the τ_d_ of the ER fluid of PIL/PANI(N_2_H_4_) is obtained by fitting the high shear rate data. Further, its τ_s_ can be used to represent the strength of suspension at the low shear rate region. As shown in Figure 6, both the τ_s_ and the τ_d_ of the suspensions of composite particles, except the τ_s_ of PIL/PANI(N_2_H_4_), are larger than that of PIL particles at 3.0 kV/mm of electric field, which indicates that the ER fluids of PIL/PANI composite particles have an enhanced ER response. For the purpose of quantifying the comparison, the relationship between yield stress and electric field strength can be fitted by the power law function $\tau \propto E^{\alpha }$. As for pure PIL, its ER response originates from local ion motion. So it is inclined to conform to the conduction model for the ER mechanism, whose α value is no more than 1.5 [43]. Here, the α value of PIL is consistent with the conduction model. While all the α values of ER fluid of PIL/PANI composite particles are more than 1.5, which indicates that the inside PANI filler make a contribution to increase polarization. Then the composite particles tend to follow the polarization model, whose α value is two at most [42]. Besides, the α values decrease gradually from PIL/PANI(HCl) to PIL/PANI(N_2_H_4_) particles, reflecting the contribution of different degrees. In addition, the τ_s_ is usually a little lager than the τ_d_, but the τ_d_ of the ER fluid of PIL/PANI(N_2_H_4_) are much different from the τ_s_. These phenomena must be related to the electro-responsive properties of PANI filler, such as intensity of polarization and dielectric polarization rate.

Figure 7 shows the typical flow curves under the electric fields of 0.0 and 3.0 kV/mm at various temperatures. Without the electric field, the off-field viscosities of all the suspensions decrease gradually with temperature increasing. This is because the viscosity of silicone oil dispersion medium decreases with temperature increasing. Under the electric field of 3.0 kV/mm, these suspensions exhibit different temperature dependence. As for the ER fluid of neat PIL particles, the flow curves are stable in a broad shear rate region, and the shear stress decreases little by little with working temperature increasing. This situation is similar to that of the ER fluid of PIL/PANI(NH_3_) particles, except that the yield stress of composite particles decreases even more. As for the ER fluid of PIL/PANI(HCl) particles, the wave trough becomes much deeper and the flow curves become more unstable. On the contrary, the flow curves of the ER fluid of PIL/PANI(N_2_H_4_) particles become more and more stable in the broad shear rate region with working temperature increasing, though there is still an upward slope below 80 °C. These differences among the ER fluids of PIL/PANI composite particles indicates that the flow stability depends on the electro-responsive properties of the PANI filler.

Figure 8 plots the temperature dependence of τ_s_ and τ_d_ under the electric field of 3.0 kV/mm. It can be clearly seen that both the τ_s_ and the τ_d_ of all the suspensions can be obtained with a wide temperature range, but tend to decrease with increasing working temperature, except for the τ_s_ of the ER fluid of PIL/PANI(N_2_H_4_) particles. Furthermore, these downward trends are different between PIL and PIL/PANI particles, which indicates that the decrease is not only related to the decrease of off-field viscosity but also the variation of the inside PANI. The yield stress of the ER fluid of PIL/PANI(HCl) particles falls sharply and its difference value between τ_s_ and τ_d_ looks very large, indicating an unstable flow behavior. The yield stress of the ER fluid of PIL/PANI(NH_3_) particles is the highest at low temperature, while it becomes near to that of PIL at high temperature. Moreover, the most obvious difference is from the ER fluid of PIL/PANI(N_2_H_4_) particles. From Figure 8B, it looks like it possesses the best temperature stability, however, it is needed to point out that its τ_d_ at 25 °C, 40 °C, and 60 °C are obtained by only fitting the high shear rate data. Its small τ_s_ indicates that there is a steep hill in the low shear rate region. Nevertheless, its τ_s_ and τ_d_ become closer and closer with temperature increasing, indicating a gradually stable flow behavior. Overall, these differences mean that the rheological behavior and temperature stability are affected by the electro-responsive properties of PANI with the temperature changes.

### 3.3. Dielectric Properties

The rheological results above clearly show that the different structural states of PANI filler have a significant influence on the flow behavior and temperature stability. In order to understand this, we investigate the dielectric properties of these suspensions due to the fact that it has been widely recognized that the polarization of particles in suspensions plays a critical role in ER response [31,44]. According to the ER mechanism, to obtain a good ER response, it is essential for suspended particles to have a large polarization intensity and a suitable polarization rate. The former influences the magnitude of interparticle interaction induced by an electric field, while the latter is closely related to the stability of interparticle interaction and the resulting flow stability under the simultaneous application of electric field and shearing field [32,42].

Figure 9 displays the angular frequency (ω) dependence of dielectric constant (ε’) and loss factor (ε”) for these ER fluids. All the fluids show an obvious dielectric relaxation peak within the measured frequency range except the ER fluid of PIL/PANI(HCl) particles. Owing to the fact that the ε’ and ε” of insulating carrier liquid are nearly independent of frequency within the measured range, the observed dielectric relaxation peak should be caused by the polarization process of dispersed particles in silicone oil. Although the ER fluids of PIL/PANI(NH_3_) and PIL/PANI(N_2_H_4_) show clear dielectric relaxations within the measured frequency range, their ε” values at the location of relaxation peak are lower than that of the ER fluid of neat PIL particles and the half-peak width of relaxation peak is also distinctly broader than that of the ER fluid of neat PIL particles. These also imply that there should be two relaxation processes attributed to outside PIL and inside PANI, respectively. Although no visible dielectric relaxation peak in the ER fluid of PIL/PANI(HCl) particles, the value of ε” increases in the high frequency region. Considering the high conductivity of PANI doping with HCl, we think there is a relaxation peak at higher frequency region from the fast polarization process of PANI filler. This can be supported by the dielectric data of previously reported composites containing high conductivity of filler inclusions [45,46]. In addition, the polarization of PIL composition in PIL/PANI(HCl) may be covered by large DC conductivity, as shown in Figure 8B. This large DC conductivity maybe because there are some PANI(HCl) particles distributed at the edge of composite like the distribution of SiO_2_ in Figure 2G–L. To fit the dielectric data and analyze the dielectric and electrical characteristics, therefore, we employ the following relaxation equation, including two Cole–Cole’s terms, a DC conductivity term, and an electrode polarization (EP) term [47,48,49]:(1)$$\varepsilon^{*}\;\left(\omega \right)=\varepsilon^{\prime }+i\varepsilon^{\prime \prime }=\varepsilon_{\infty }^{\prime }+\frac{\Delta \varepsilon_{1}^{\prime }}{1+{\left(i\omega \lambda_{1}\right)}^{\beta_{1}}}+\frac{\Delta \varepsilon_{2}^{\prime }}{1+{\left(i\omega \lambda_{2}\right)}^{\beta_{2}}}+i\frac{\sigma }{\varepsilon_{0}\omega }+A\omega^{-n}$$

The equations of **ε**′ and **ε**″ are shown below, where subscript 1 for outside PIL matrix and subscript 2 for inside PANI filler:(2)$$\varepsilon^{\prime }=\varepsilon_{\infty }^{\prime }+\Delta \varepsilon_{1}^{\prime }\left(\frac{1+{\left(\omega \lambda_{1}\right)}^{\beta_{1}}\cos\left(\frac{\pi \beta_{1}}{2}\right)}{1+2{\left(\omega \lambda_{1}\right)}^{\beta_{1}}\cos\left(\frac{\pi \beta_{1}}{2}\right)+{\left(\omega \lambda_{1}\right)}^{2\beta_{1}}}\right)+\Delta \varepsilon_{2}^{\prime }\left(\frac{1+{\left(\omega \lambda_{2}\right)}^{\beta_{2}}\cos\left(\frac{\pi \beta_{2}}{2}\right)}{1+2{\left(\omega \lambda_{2}\right)}^{\beta_{2}}\cos\left(\frac{\pi \beta_{2}}{2}\right)+{\left(\omega \lambda_{2}\right)}^{2\beta_{2}}}\right)+A\omega^{-n}$$
(3)$$\varepsilon^{\prime \prime }=\Delta \varepsilon_{1}^{\prime }\left(\frac{{\left({\omega \lambda }_{1}\right)}^{\beta_{1}}\sin\left(\frac{{\pi \beta }_{1}}{2}\right)}{1+2{\left({\omega \lambda }_{1}\right)}^{\beta_{1}}\cos\left(\frac{{\pi \beta }_{1}}{2}\right)+{\left({\omega \lambda }_{1}\right)}^{2\beta_{1}}}\right)+\Delta \varepsilon_{2}^{\prime }\left(\frac{{\left({\omega \lambda }_{2}\right)}^{\beta_{2}}\sin\left(\frac{{\pi \beta }_{2}}{2}\right)}{1+2{\left({\omega \lambda }_{2}\right)}^{\beta_{2}}\cos\left(\frac{{\pi \beta }_{2}}{2}\right)+{\left({\omega \lambda }_{2}\right)}^{2\beta_{2}}}\right)+\frac{\sigma }{\varepsilon_{0}\omega }$$
where $\Delta \varepsilon^{\prime }=\varepsilon_{0}^{\prime }-\varepsilon_{\infty }^{\prime }$ is the dielectric relaxation strength ($\varepsilon_{0}^{\prime }$ and $\varepsilon_{\infty }^{\prime }$ are the limit values of the relative dielectric constant at the frequencies below and above the relaxation frequencies, respectively), $\lambda =1/\omega_{\max}$ is the dielectric relaxation time ($\omega_{\max}$ is the local angular frequency of the dielectric loss peak), β (0 < β ≤ 1) is the Cole–Cole parameter leading to a symmetrical broadening for the spectrum, σ is the DC conductivity, *A* is related to the amplitude of EP, and *n* is related to the slope of EP’s high frequency tail.

As shown in Figure 9, Equations (2) and (3) can well fit the experimental dielectric data and Table 1 summarizes the dielectric characteristics. It is found that two relaxation processes in composites can be clarified, which can be respectively attributed to PIL matrix and PANI filler according to the magnitude of λ, and the conductivity of different forms of PANI after doping or dedoping. However, the magnitude of deviation between two relaxation times is different among these composites. The value of λ of PANI(NH_3_) filler is close to that of PIL matrix, while the value of λ of PANI(HCl) filler is far from that of PIL matrix. Compared it with the flow curves in Figure 5, the smaller the magnitude of deviation between two relaxation times is, the more stable the flow curves are. Among these suspensions, the flow curves of the ER fluid of PIL/PANI(NH_3_) composite particles are most stable in the broad shear rate range (see Figure 5C). According to the proposal ER mechanism, the polarization rate of particles is important to the stability of the flow curve of ER suspensions under the simultaneous effects of both electric and shearing fields. Under DC electric fields, it is proposed that ER fluids with a relaxation time of 1.6 × 10^−2^–1.6 × 10^−6^ s are appropriate for achieving a stable flow curve. Too slow or too fast a relaxation time is not favorable to the stability during flow because then too slow a relaxation time easily results in insufficient particle polarization during flow, while too fast a relaxation time easily results in an increase of repulsive interaction between particles due to the difference between the polarization direction and the direction connecting two particles [31]. Here, the λ of PIL matrix in composites is located in 1.6 × 10^−2^–1.6 × 10^−6^ s but the λ of PANI filler is located in different regions. The λ of PANI filler in PIL/PANI(N_2_H_4_) is relatively slow, which slows down the polarization rate of particles leading to insufficient interparticle interaction. The λ of PANI filler in PIL/PANI(HCl) is too fast, which can result in an increase of repulsive interaction between particles. This extreme mismatching can cause that the polarization of PANI(HCl) filler and PIL matrix offsets each other at a high shear rate region. As a consequence, this counteraction leads to the decline of interparticle interaction, which further results in the decline of shear stress. So, there is a deep wave trough at the high shear rate under high electric filed (see Figure 5B). However, the **λ** of PANI filler in PIL/PANI(NH_3_) is close to the **λ** of PIL matrix, which results in stable flow curves. Moreover, the **λ** values of PIL/PANI composite particles are not consistent with that of PIL by using the same preparation process. This suggests that there is a synergistic effect like electrostatic interaction between outside PIL and inside PANI, or the post processing could affect the outside PIL such as the high-temperature reaction that changes the surface of PIL/PANI(N_2_H_4_) particles (see Figure 1D). In addition, we note that the total value of ∆**ε**’ of the ER fluid of PIL/PANI is not much larger than that of the ER fluid of neat PIL, which is not enough to explain the enhanced yield stress. This enhancement due to the improvement of ER structure has been investigated in our previous work, and, therefore, is not discussed here [38]. Compared to other PIL-based composite, like GO/PPy/PIL composite nanosheets or PIL/SiO_2_ composite particles [29,30], the ∆ε’ of PIL/PANI(NH_3_) composites is also relatively larger and λ is more suitable and thus result in enhanced ER response compared to them.

Figure 10 shows the dielectric relaxation spectra of these ER fluids at various temperatures. With temperature increasing, the dielectric relaxation peak of all these fluids shifts to high **ω**, which means the increase of polarization rate, and the **ε**’ and **ε**” values rise up quickly in the low **ω** region duo to electrode polarization and DC conductivity. We note that the ER fluid of PIL/PANI(HCl) particles also has electrode polarization owing to the quick increase of **ε**’ (see Figure 10B), which indicates that there are ions moving to the electrode, and the relaxation peaks of PANI for PIL/PANI(NH_3_) and PIL/PANI(N_2_H_4_) composite particles are covered up by the high conductivity of PIL (see Figure 10C,D). These are further signs that PIL is on the outside. Compared to others, we also note that the relaxation peak of the ER fluid of PIL/PANI(NH_3_) becomes wider with temperature increasing, which also indicates that there are two relaxation processes for the composite particles.

By using Equations (2) and (3), we can obtain the dielectric relaxation time of PIL matrix and PANI filler. Figure 11A shows the temperature dependence of the reciprocal of their dielectric relaxation time (λ^−1^). Owing to the fact that the relaxation peak shift to high ω is thermally promoted, the λ^−1^ can follow the Arrhenius equation:(4)$$\lambda^{-1}\propto e^{-E_{a}/RT}$$
where *E*_a_ is the activation energy, *R* is the molar gas constant, and *T* is the absolute temperature. The *E*_a_ value can be calculated by the slope on the 1000/*T* coordinate. This Arrhenius-type temperature dependence of *E*_a_ for PIL matrix indicates that the ion motion of PIL is from the thermal diffusion, and not coupled with the chain segment [50,51]. What is more, it can be found that the *E*_a_ of PIL for PIL/PANI composite particles is slightly larger than that of pure PIL. This may be because that after wrapping the PANI, the interfaces between PIL and PANI can hinder the ion motion of mobile anion leading to the increase of *E*_a_ of PIL. As for PANI, the *E*_a_ corresponds to the energy gap of semiconducting polymer, which is determined by its band structure. The smaller the energy gap of PANI is, the better the conductivity is. The PANI doping with HCl is close to metallicity, so it possesses the smallest *E*_a_. Compared to the PANI dedoping with NH_3_, many quinoid units of the PANI reduced by N_2_H_4_ transform into benzene rings. This change can increase the energy gap resulting in the largest *E*_a_ for PANI(N_2_H_4_) [35,40].

The differences in *E*_a_ values between PIL matrix and PANI filler for composite particles can bring more differences in dielectric polarization rate with increasing temperature. Figure 11B plots the variation of the ratios of *λ*_PIL_ of PIL matrix to *λ*_PANI_ of PANI filler with temperature. This ratio can reflect the magnitude of difference of two polarization rates. From Figure 11B, it can be seen that the ratio of PIL/PANI(HCl) is much larger than others, which exhibits a huge mismatching between PIL matrix and PANI(HCl) filler. As a result, the flow curves are not stable within the measured temperature (see Figure 7B). As for PIL/PANI(NH_3_), the ratio is closest to one at room temperature, so it possesses the most stable flow curves and the largest ***τ***_d_ at this condition. However, the ratio starts to depart from one with increasing temperature, indicating the increase of the mismatching. This change leads to the rapid decline of ***τ***_d_ (see Figure 8). On the contrary, the ratio of PIL/PANI(N_2_H_4_) becomes closer to one as temperature increases, which leads to more and more stable flow behavior (see Figure 7D). Therefore, the temperature modulated rheological tests further support the conclusion that the flow behavior of the ER fluids of PIL/PANI composite particles depends on the matching degree of dielectric polarization rate between PIL matrix and PANI filler. The closer their polarization rates are, the more stable flow curve is in a broad shear rate region under an external electric field.

## 4. Conclusions

By using different structural states of semiconducting PANI as filler, we investigate the influence of polarization rate difference of filler and matrix on the flow behavior of electrorheological fluids of PIL/PANI composite particles under external electric fields. The PIL/PANI composite particles have been firstly prepared by a post counterion conversion procedure, and then treated by ammonia or hydrazine to achieve different electrical responses of PANI filler. Under electric fields, only the PIL/PANI(NH_3_) particles show a stable flow behavior in a broad shear rate region and an enhanced ER response at room temperature. Nevertheless, the flow curves of PIL/PANI(N_2_H_4_) become more and more stable with measure temperature increasing. By using dielectric spectroscopy, it can be found that the polarization rate of matrix should be located in an appropriate range, and then the flow stability depends on how closely the dielectric polarization rates of the outside PIL matrix and inside PANI filler match. The closer they are, the more stable the flow curves are. This result may provide guidance to select suitable materials with matched polarization rates as matrix and filler to design optimal ER composite materials with enhanced and stable ER effect. In order to fix this conclusion, more studies, such as core-shell-like and Janus-like ER particles, are worth further investigation because of their tunable morphology and property.

## Figures and Tables

**Figure 1 polymers-12-00703-f001:**
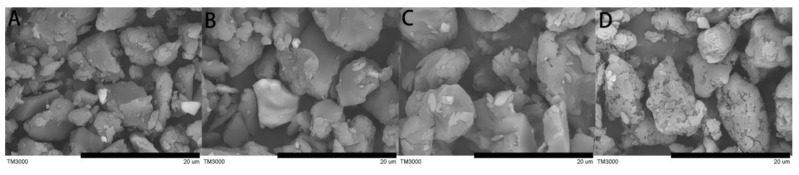
SEM images of samples: (**A**) poly(ionic liquid) (PIL), (**B**) PIL/polyaniline (PANI)(HCl), (**C**) PIL/PANI(NH_3_), and (**D**) PIL/PANI(N_2_H_4_) (Scale bar = 20 μm).

**Figure 2 polymers-12-00703-f002:**
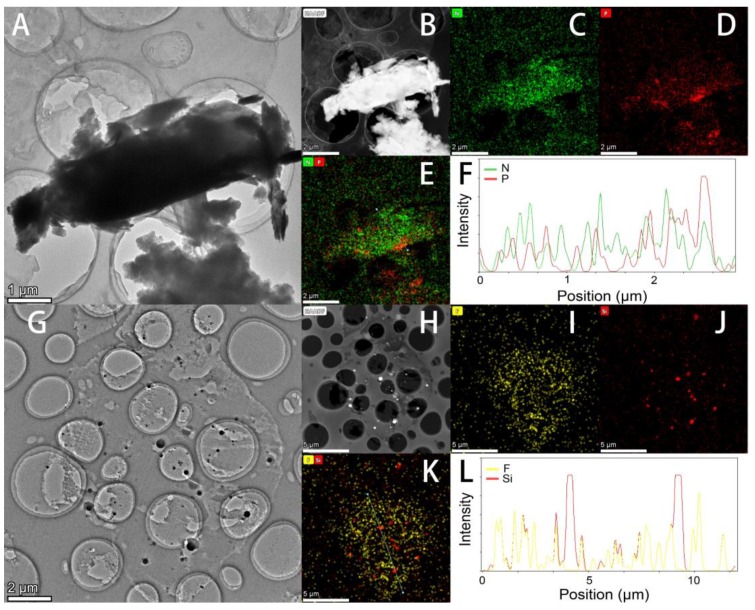
TEM images, HAADF images, and linear EDS plots of ultrathin cross-sections on holey carbon films: (**A**,**B**,**F**) PIL/PANI and (**G**,**H**,**L**) PIL/SiO_2_. EDS elemental maps: (**C**) N, (**D**) P, (**E**) N and P, (**I**) F, (**J**) Si, and (**K**) F and Si. (**B**–**E** scale bar = 2 μm, **H**–**K** scale bar = 5 μm)

**Figure 3 polymers-12-00703-f003:**
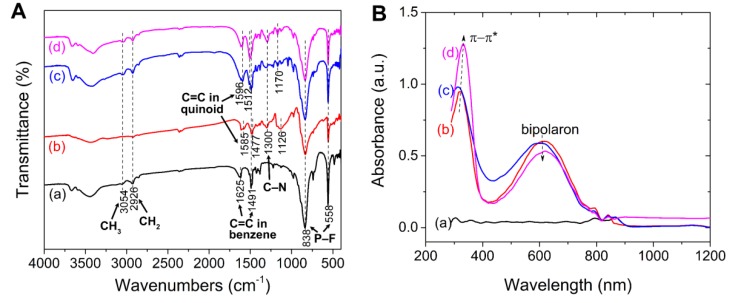
(**A**) FT-IR spectra and (**B**) UV-vis spectra of samples: (**a**) neat PIL, (**b**) PIL/PANI(HCl), (**c**) PIL/PANI(NH_3_), and (**d**) PIL/PANI(N_2_H_4_).

**Figure 4 polymers-12-00703-f004:**
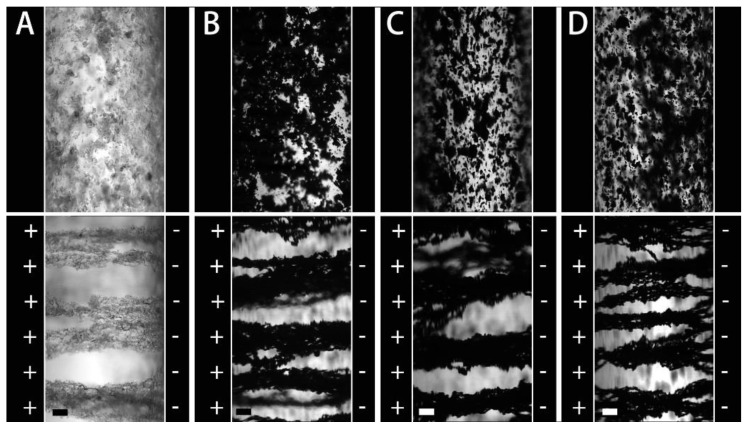
Optical photographs of the electrorheological (ER) fluids without and with electric field: (**A**) neat PIL, (**B**) PIL/PANI(HCl), (**C**) PIL/PANI(NH_3_), and (**D**) PIL/PANI(N_2_H_4_) (*ϕ* = 3 vol%, Scale bar = 100 μm).

**Figure 5 polymers-12-00703-f005:**
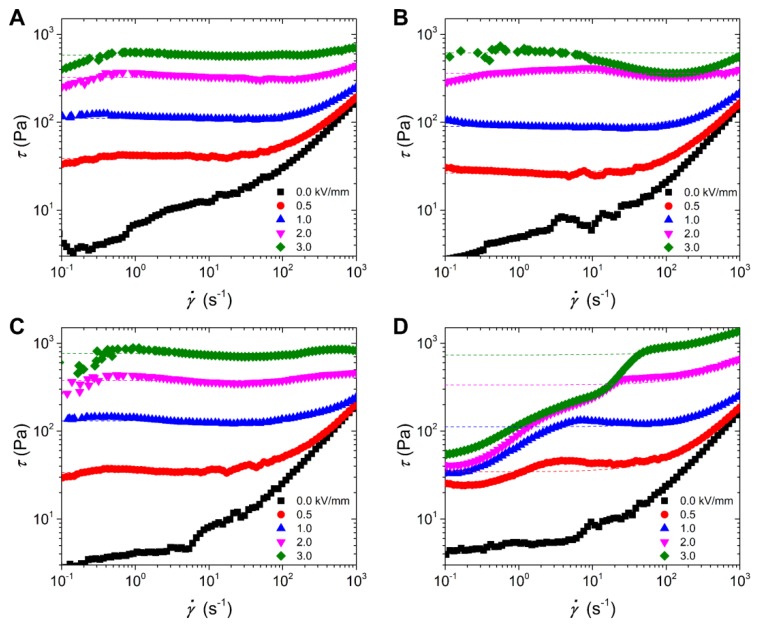
Flow curves of shear stress vs. shear rate for the ER fluids: (**A**) neat PIL, (**B**) PIL/PANI(HCl), (**C**) PIL/PANI(NH_3_), and (**D**) PIL/PANI(N_2_H_4_) (*T* = 25 °C).

**Figure 6 polymers-12-00703-f006:**
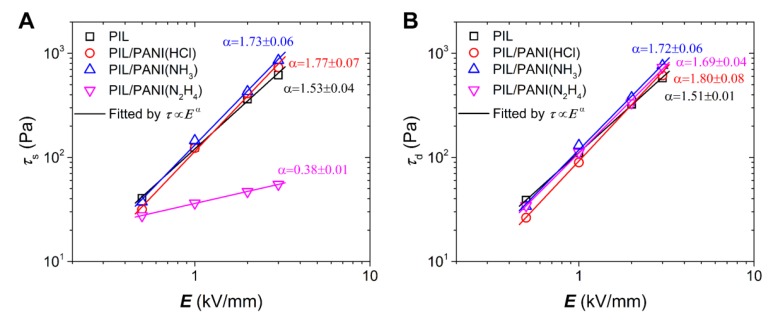
(**A**) Static yield stress and (**B**) dynamic yield stress as a function of electric field strength for the ER fluids (*T* = 25 °C).

**Figure 7 polymers-12-00703-f007:**
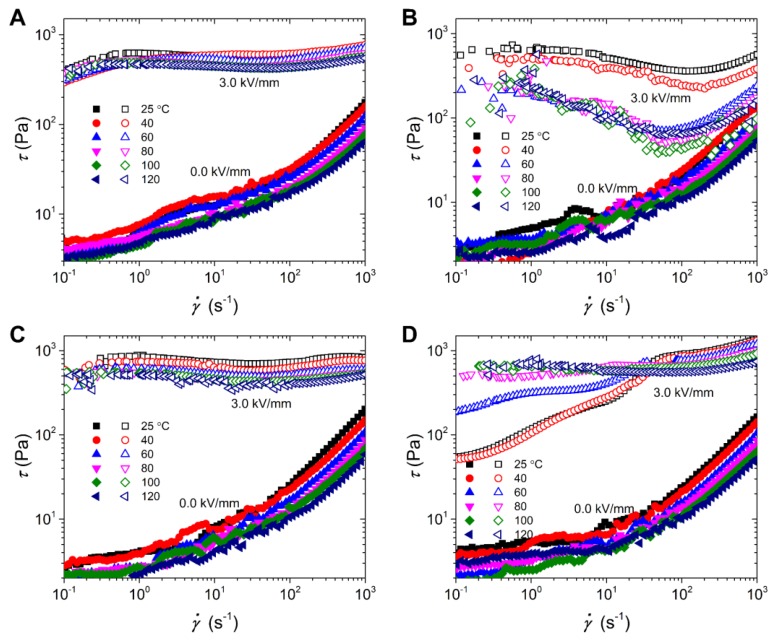
Flow curves of shear stress vs. shear rate for the ER fluids at various temperatures: (**A**) neat PIL, (**B**) PIL/PANI(HCl), (**C**) PIL/PANI(NH_3_), and (**D**) PIL/PANI(N_2_H_4_).

**Figure 8 polymers-12-00703-f008:**
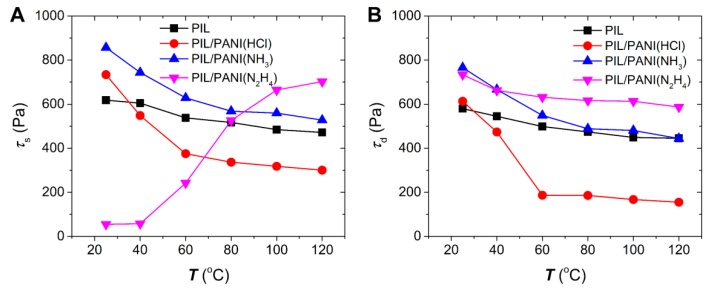
(**A**) Static yield stress and (**B**) dynamic yield stress at 3.0 kV/mm plotted as a function of temperature for the ER fluids of PIL/PANI particles.

**Figure 9 polymers-12-00703-f009:**
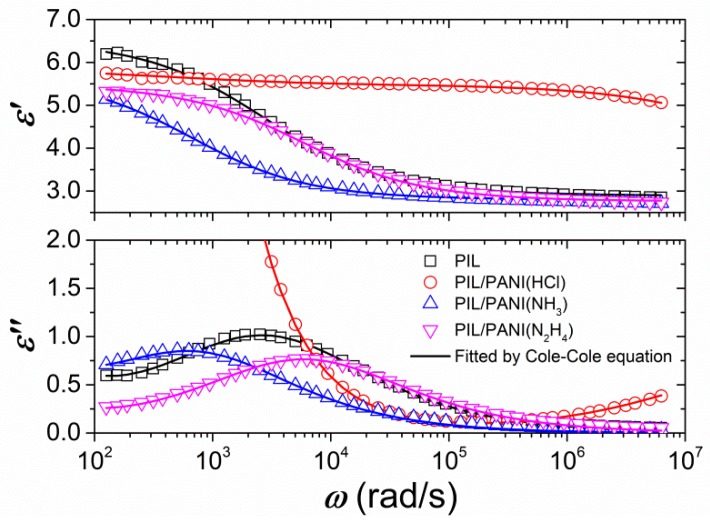
Dielectric relaxation spectra of the ER fluids (*T* = 30 °C). Solid lines show the best fit of ***ε***′ and ***ε***″ by Equations (2) and (3).

**Figure 10 polymers-12-00703-f010:**
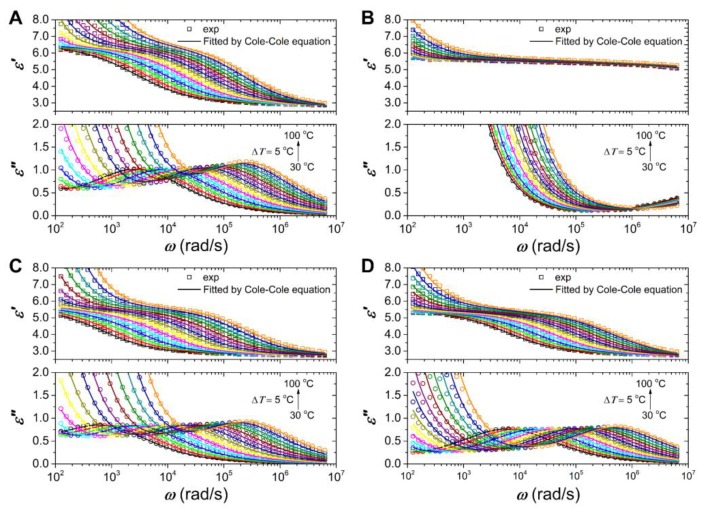
Dielectric relaxation spectra of the ER fluids at various temperatures: (**A**) neat PIL, (**B**) PIL/PANI(HCl), (**C**) PIL/PANI(NH_3_), and (**D**) PIL/PANI(N_2_H_4_).

**Figure 11 polymers-12-00703-f011:**
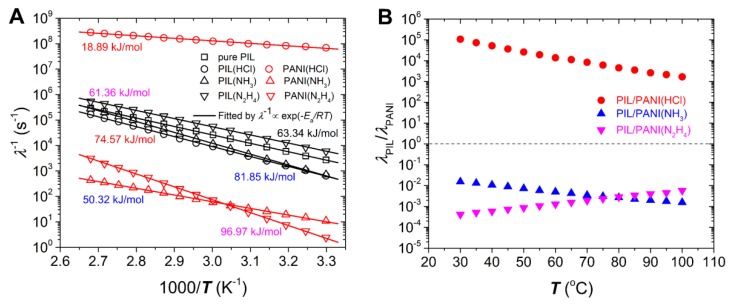
Temperature dependence of (**A**) *λ*^−1^ and (**B**) *λ*_PIL_/*λ*_PANI_.

**Table 1 polymers-12-00703-t001:** Dielectric characteristics of the ER fluids of PIL/PANI. (*T* = 30 °C)

Sample	*ε*’_∞_	∆*ε*’ ^a^		*λ* (s)		*σ* (S/m)
		∆*ε*’_1_	∆*ε*’_2_	*λ* _1_	*λ* _2_	
PIL	2.89	3.65		3.7 × 10^-4^		~2.48 × 10^−10^
PIL/PANI(HCl)	2.78	0.32	2.7	1.55 × 10^−3^	1.45 × 10^−8^	~4.78 × 10^−8^
PIL/PANI(NH_3_)	2.80	2.78	0.8	1.40 × 10^−3^	9.10 × 10^−2^	~7.97 × 10^−11^
PIL/PANI(N_2_H_4_)	2.76	2.72	0.4	1.68 × 10^−4^	4.10 × 10^−1^	~7.08 × 10^−11^

^a^ Subscript 1 for outside PIL and subscript 2 for inside PANI.

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
