# Peer review of "The Effect of Dielectric Polarization Rate Difference of Filler and Matrix on the Electrorheological Responses of Poly(ionic liquid)/Polyaniline Composite Particles"

_polymers, 2020, doi:10.3390/polym12030703_

Round 1
Reviewer 1 Report
Manuscript is in good shape now, Please accept for publication.
Author Response
Dear Reviewer:
Many thanks for your positive recommending about our manuscript (Manuscript ID: polymers-747603; Title: The effect of dielectric polarization rate difference of filler and matrix on the electrorheological responses of poly(ionic liquid)/polyaniline composite particles.) The comments will undoubtedly improve the quality of our paper. We have finished the revision accordingly. The detailed responses and changes (highlighted in the revised manuscript by Blue) are listed in the attached file.

Reviewer 2 Report
The paper was previous reviewed by two reviewers, and I think that the authors answered well. And the paper can be accepted. Only two comments: For the caption of Figures 5-8 and 10, the Ï•=20 vol% is unnecessary. For the Figure 11A, the slope is the activation energy. And what is the mean for slope in Figure 11B
Author Response
Dear Reviewer:
Many thanks for your positive recommending and valuable comments about our manuscript (Manuscript ID: polymers-747603; Title: The effect of dielectric polarization rate difference of filler and matrix on the electrorheological responses of poly(ionic liquid)/polyaniline composite particles.) These comments will undoubtedly improve the quality of our paper. We have finished revision accordingly. The detailed responses and changes (highlighted in revised manuscript by Blue) are listed in the attached file.

This manuscript is a resubmission of an earlier submission. The following is a list of the peer review reports and author responses from that submission.
Round 1
Reviewer 1 Report
Manuscript: The effect of dielectric polarization rate difference of filler and matrix on the electrorheological responses of poly(ionic liquid)/polyaniline composite particles
polymers-702835
The manuscript presents very good research work related to dielectric polarization properties of poly(ionic liquid)/polyaniline composite.
Some minor comments are as follows;
Authors need to include some interesting data in the abstract part of the manuscript. Authors need incorporate recent references related to the subject in the manuscript. Figure 2 authors need to label the functional groups in the FT-IR. Authors need to compare the dielectric constant results with previously reported similar materials. Future prospective of the presented research need to incorporate in the conclusion part of the manuscript.
Author Response
Dear Reviewer:
Many thanks for your constructive criticisms and valuable comments about our manuscript (Manuscript ID: polymers-702835; Title: The effect of dielectric polarization rate difference of filler and matrix on the electrorheological responses of poly(ionic liquid)/polyaniline composite particles.) These criticisms and comments will undoubtedly improve the quality of our paper. We have finished the revision accordingly. The detailed responses and changes (highlighted in the revised manuscript by Blue) are listed in the attached file.

Reviewer 2 Report
General comments
Manucript is mundane, nothing new here. Infact the work is similar to 10.1039/C4TA00828F, 10.1039/C6SM02480G etc. and the present manuscript is only an incremental change both in terms of knowledge and in data. If published this manuscript will have no or minimal impact. Even in his mediocre incremental work, most references are irrelevant. Authors need to thoroughly revamp their references but before that they are strongly advised to read this paper "Cite with a Sight" J. Phys. Chem. Lett. 2014, 5, 7, 1241-1242.
Specific comments
Line #33.. Authors state "show obvious viscosity increase".. it's not OBVIOUS.. there are many scenarios wherein viscosity decreases under external electric field.
Lines #43-44: This statement implies that there is relation ship between 'texture' and ER activity. But the cited refe #16 doesn't even mention the term "TEXTURE" in the whole manuscript. Give appropriate reference.
Lines #47 -48: Authors usage of the term "intrinsically hydrophobic nature of constituent ions" is highly misleading. What ever fancy definition you may give to Ionic liquids (IL), but at the end of the day IL are salts and most ILs (if not all) are miscible in water. Ex: 10.3390/ijms10031271; 10.1039/C6RA06791C. In this context, the cited references #19 and #20 are irrelevant. Replace them with appropriate references.
Results and Discussion
Quality of TEM images is abysmal. Give individual maps N, P, C, O etc.. and a combined composite map. Secondly no HAADF image which is also must. Authors are strongly advised to read ISO 22309 standard and report their TEM EDS data accordin to those standards. In revised manuscript besides EDS map, EDS plot must also be included.
Authors mentioned "HRTEM equipped with energy dispersive X-ray spectroscopy (EDX)." what was the operating voltage? What EDS analyser was used in the tests?
Author Response

(The authors gave the same response as above.)

Round 2
Reviewer 2 Report
As stated earlier by me this manuscript is mundane, nothing new here. Infact the work is similar to 10.1039/C4TA00828F, 10.1039/C6SM02480G etc. and the present manuscript is only an incremental change both in terms of knowledge and in data. The changes and modifications in this revised manuscript are cosmetic and did not improve the quality of the manuscript. If published this manuscript will have no or minimal impact.